# Unposed Sparse Views Room Layout Reconstruction in the Age of Pretrain Model

**Yaxuan Huang**[1]*    **Xili Dai**[2]*    **Jianan Wang**[3]    **Xianbiao Qi**[4]    **Yixing Yuan**[1]
**Xiangyu Yue**[5]†

[1]Hong Kong Center for Construction Robotics, The Hong Kong University of Science and Technology
[2]The Hong Kong University of Science and Technology (Guangzhou)
[3]Astribot  [4]Intellifusion Inc.
[5]Multimedia Lab (MMLab) and SHIAE, The Chinese University of Hong Kong

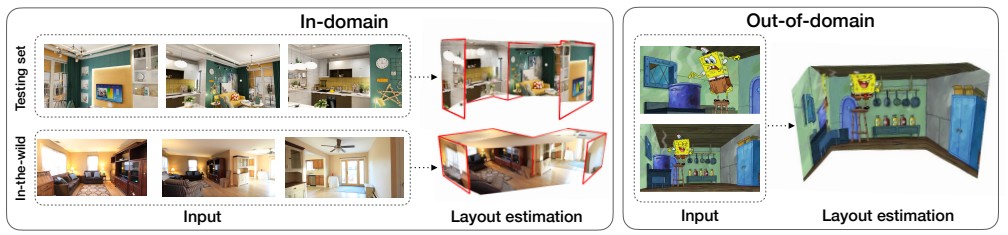

Figure 1: We present a novel method for estimating room layouts from a set of unconstrained indoor images. Our approach demonstrates robust generalization capabilities, performing well on both in-the-wild datasets (Zhou et al., 2018) and out-of-domain cartoon (Weber et al., 2024) data.

## Abstract

Room layout estimation from multiple-perspective images is poorly investigated due to the complexities that emerge from multi-view geometry, which requires muti-step solutions such as camera intrinsic and extrinsic estimation, image matching, and triangulation. However, in 3D reconstruction, the advancement of recent 3D foundation models such as DUSt3R has shifted the paradigm from the traditional multi-step structure-from-motion process to an end-to-end single-step approach. To this end, we introduce Plane-DUSt3R, a novel method for multi-view room layout estimation leveraging the 3D foundation model DUSt3R. Plane-DUSt3R incorporates the DUSt3R framework and fine-tunes on a room layout dataset (Structure3D) with a modified objective to estimate structural planes. By generating uniform and parsimonious results, Plane-DUSt3R enables room layout estimation with only a single post-processing step and 2D detection results. Unlike previous methods that rely on single-perspective or panorama image, Plane-DUSt3R extends the setting to handle multiple-perspective images. Moreover, it offers a streamlined, end-to-end solution that simplifies the process and reduces error accumulation. Experimental results demonstrate that Plane-DUSt3R not only outperforms state-of-the-art methods on the synthetic dataset but also proves robust and effective on in the wild data with different image styles such as cartoon. Our code is available at: https://github.com/justacar/Plane-DUSt3R

## 1 Introduction

3D room layout estimation aims to predict the overall spatial structure of indoor scenes, playing a crucial role in understanding 3D indoor scenes and supporting a wide range of applications. For example, room layouts could serve as a reference for aligning and connecting other objects in indoor environment reconstruction (Nie et al., 2020). Accurate layout estimation also aids robotic path planning and navigation by identifying passable areas (Mirowski et al., 2016). Additionally, room

---

* Equal contribution, † Corresponding author

layouts are essential in tasks such as augmented reality (AR) where spatial understanding is critical. Therefore, 3D room layout estimation has attracted considerable research attention with continued development of datasets (Zheng et al., 2020; Wang et al., 2022) and methods (Yang et al., 2022; Stekovic et al., 2020; Wang et al., 2022) over the past few decades.

Methods for 3D room layout estimation (Zhang et al., 2015; Hedau et al., 2009; Yang et al., 2019) initially relied on the Manhattan assumption with a single perspective or panorama image as input. Over time, advancements (Stekovic et al., 2020) have relaxed the Manhattan assumption to accommodate more complex settings, such as the Atlanta model, or even no geometric assumption at all. Recently, Wang et al. (2022) introduced a "multi-view" approach, capturing a single room with two panorama images, marking the first attempt to extend the input from a single image to multiple images. Despite this progress, exploration in this direction remains limited, hindered by the lack of well-annotated multi-view 3D room layout estimation dataset.

Currently, multi-view datasets with layout annotations are very scarce. Even the few existing datasets, such as Structure3D (Zheng et al., 2020), provide only a small number of perspective views (typically ranging from 2 to 5). This scarcity of observable views highlights a critical issue: wide-baseline sparse-view structure from motion (SfM) remains an open problem. Most contemporary multi-view methods (Wang et al., 2022; Hu et al., 2022) assume known camera poses or start with noisy camera pose estimates. Therefore, solving wide-baseline sparse-view SfM would significantly advance the field of multi-view 3D room layout estimation. The recent development of large-scale training and improved model architecture offers a potential solution. While GPT-3 (Brown, 2020) and Sora (Brooks et al., 2024) have revolutionized NLP and video generation, DUSt3R (Wang et al., 2024) brings a paradigm shift for multi-view 3D reconstruction, transitioning from a multi-step SfM process to an end-to-end approach. DUSt3R demonstrates the ability to reconstruct scenes from unposed images, without camera intrinsic/extrinsic or even view overlap. For example, with two unposed, potentially non-overlapping views, DUSt3R could generate a 3D pointmap while inferring reasonable camera intrinsic and extrinsic, providing an ideal solution to the challenges posed by wide-baseline sparse-view SfM in multi-view 3D room layout estimation.

In this paper, we employ DUSt3R to tackle the multi-view 3D room layout estimation task. Most single-view layout estimation methods (Yang et al., 2022) follow a two-step process: 1) extracting 2D & 3D information, and 2) lifting the results to a 3D layout with layout priors. When extending this approach to multi-view settings, an additional step is required: establishing geometric primitive correspondence across multi-view before the 3D lifting step. Given the limited number of views in existing multi-view layout datasets, this correspondence-establishing step essentially becomes a sparse-view SfM problem. Hence, incorporating a single-view layout estimation method with DUSt3R to handle multi-view layout estimation is a natural approach. However, this may introduce a challenge: independent plane normal estimation for each image fails to leverage shared information across views, potentially reducing generalizability to unseen data in the wild. To this end, we adopt DUSt3R to solve correspondence establishement and 3D lifting simultaneously, which jointly predict plane normal and lift 2D detection results to 3D. Specifically, we modify DUSt3R to estimate room layouts directly through dense 3D point representation (pointmap), focusing exclusively on structural surfaces while ignoring occlusions. This is achieved by retraining DUSt3R with the objective to predict only structural planes, the resulting model is named Plane-DUSt3R. However, dense pointmap representation is redundant for room layout, as a plane can be efficiently represented by its normal and offset rather than a large number of 3D points, which may consume significant space. To streamline the process, we leverage well-established off-the-shelf 2D plane detector to guide the extraction of plane parameters from the pointmap. We then apply post-processing to obtain plane correspondences across different images and derive their adjacency relationships.

Compared to existing room layout estimation methods, our approach introduces the first pipeline capable of unposed multi-view (perspective images) layout estimation. Our contributions can be summarized as follows:

1. We propose an unposed multi-view (sparse-view) room layout estimation pipeline. To the best of our knowledge, this is the first attempt at addressing this natural yet underexplored setting in room layout estimation.

2. The introduced pipeline consists of three parts: 1) a 2D plane detector, 2) a 3D information prediction and correspondence establishment method, Plane-DUSt3R, and 3) a post-processing algorithm. The 2D detector was retrained with SOTA results on the Structure3D dataset (see Ta-

ble 3). The Plane-DUSt3R achieves a $5.27\%$ and $5.33\%$ improvement in RRA and mAA metrics, for the multi-view correspondence task compared to state-of-the-art methods (see Table 2).

3. In this novel setting, we also design several baseline methods for comparison to validate the advantages of our pipeline. Specifically, we outperform the baselines by 4 projection 2D metrics and 1 3D metric respectively (see Table 1). Furthermore, our pipeline not only performs well on the Structure3D dataset (see Figure 6), but also generalizes effectively to in-the-wild datasets (Zhou et al., 2018) and scenarios with different image styles such as cartoon style (see Figure 1).

## 2 RELATED WORK

**Layout estimation.** Most room layout estimation research focuses on single-perspective image inputs. Stekovic et al. (2020) formulates layout estimation as a constrained discrete optimization problem to identify 3D polygons. Yang et al. (2022) introduces line-plane constraints and connectivity relations between planes for layout estimation, while Sun et al. (2019) formulates the task as predicting 1D layouts. Other studies, such as Zou et al. (2018), propose to utilize monocular 360-degree panoramic images for more information. Several works extend the input setting from single panoramic to multi-view panoramic images, *e.g.* Wang et al. (2022) and Hu et al. (2022). However, there is limited research addressing layout estimation from multi-view RGB perspective images. Howard-Jenkins et al. (2019) detects and regresses 3D piece-wise planar surfaces from a series of images and clusters them to obtain the final layout, but this method requires posed images. The most related work is Jin et al. (2021), which focuses on a different task: reconstructing indoor scenes with planar surfaces from wide-baseline, unposed images. It is limited to two views and requires an incremental stitching process to incorporate additional views.

**Holistic scene understanding.** Traditional 3D indoor reconstruction methods are widely applicable but often lack explicit semantic information. To address this limitation, recent research has increasingly focused on incorporating holistic scene structure information, enhancing scene understanding by improving reasoning about physical properties, mostly centered on single-perspective images. Several studies have explored the detection of 2D line segments using learning-based detectors (Zhou et al., 2019; Pautrat et al., 2021; Dai et al., 2022). However, these approaches often struggle to differentiate between texture-based lines and structural lines formed by intersecting planes. Some research has focused on planar reconstruction to capture higher-level information (Liu et al., 2018; Yu et al., 2019; Liu et al., 2019). Certain studies (Huang et al., 2018; Nie et al., 2020; Sun et al., 2021) have tackled multiple tasks alongside layout reconstruction, such as depth estimation, object detection, and semantic segmentation. Other works operate on constructed point maps; for instance, Yue et al. (2023) reconstructs floor plans from density maps by predicting sequences of room corners to form polygons. SceneScript (Avetisyan et al., 2024) employs large language models to represent indoor scenes as structured language commands.

**Multi-view pose estimation and reconstruction.** The most widely applied pipeline for pose estimation and reconstruction on a series of images involves SfM (Schönberger & Frahm, 2016) and MVS (Schönberger et al., 2016), which typically includes steps such as feature mapping, finding correspondences, solving triangulations and optimizing camera parameters. Most mainstream methods build upon this paradigm with improvements on various aspects of the pipeline. However, recent works such as DUSt3R (Wang et al., 2024) and MASt3R (Leroy et al., 2024) propose a reconstruction pipeline capable of producing globally-aligned pointmaps from unconstrained images. This is achieved by casting the reconstruction problem as a regression of pointmaps, significantly relaxing input requirements and establishing a simpler end-to-end paradigm for 3D reconstruction.

## 3 METHOD

In this section, we formulate the layout estimation task, transitioning from a single-view to a multi-view scenario. We then derive our multi-view layout estimation pipeline as shown in Figure 2 (Section 3.1). Our pipeline consists of three parts: a 2D plane detector $f_1$, a 3D information prediction and correspondence establishment method Plane-DUSt3R $f_2$ (Section 3.2), and a post-processing algorithm $f_3$ (Section 3.3).

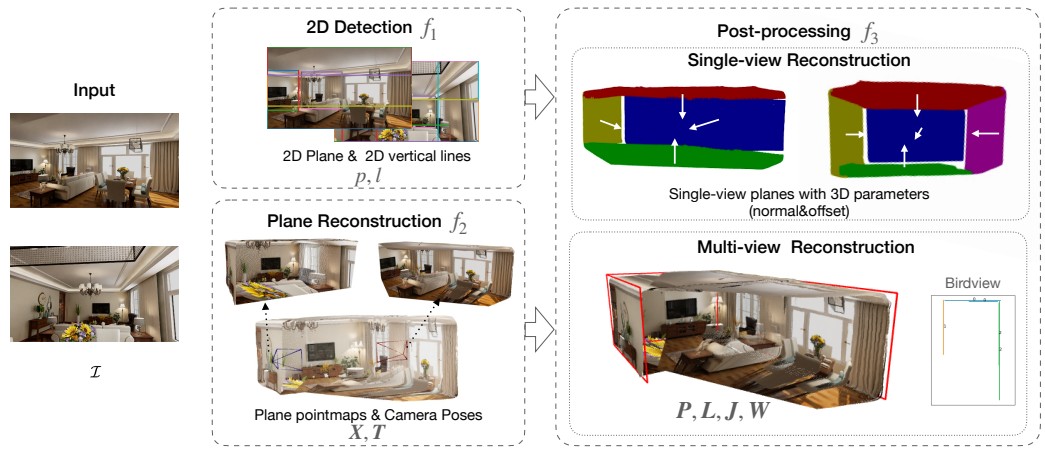

Figure 2: Our multi-view room layout estimation pipeline. It consists of three parts: 1) a 2D plane detector $f_1$, 2) a 3D information prediction and correspondence establishment method Plane-DUSt3R $f_2$, and 3) a post-processing algorithm $f_3$.

## 3.1 FORMULATION OF THE MULTI-VIEW LAYOUT ESTIMATION TASK

We begin by revisiting the single-view layout estimation task and unifying the formulation of existing methods. Next, we extend the formulation from single-view to multiple-view setting, providing a detailed analysis and discussion focusing on the choice of solutions. Before formulating the layout estimation task, we adopt the "geometric primitives + relationships" representation from Zheng et al. (2020) to model the room layout.

**Geometric Primitives.**

- **Planes:** The scene layout could be represented as a set of planes $\{P_1, P_2 \ldots\}$ in 3D space and their corresponding 2D projections $\{p_1, p_2, \ldots\}$ in images. Each plane is parameterized by its normal $n \in \mathbb{S}^2$ and offset $d$. For a 3D point $x \in \mathbb{R}^3$ lying on the plane, we have $n^T x + d = 0$.
- **Lines & Junction Points:** In 3D space, two planes intersect at a 3D line, three planes intersect at a 3D junction point. We denote the set of all 3D lines/junction points in the scene as $\{L_1, L_2 \ldots\}/\{J_1, J_2 \ldots\}$ and their corresponding 2D projections as $\{l_1, l_2, \ldots\}/\{j_1, j_2, \ldots\}$ in images.

**Relationships.**

- **Plane/Line relationships:** An adjacent matrix $W_p/W_l \in \{0, 1\}$ is used to model the relationship between planes/lines. Specifically, $W_p(i, j) = 1$ if and only if $P_i$ and $P_j$ intersect along a line; otherwise, $W_p(i, j) = 0$. Similarly to plane relationship, $W_l(i, j) = 1$ if and only if $L_i$ and $L_j$ intersect at a certain junction, otherwise, $W_l(i, j) = 0$.

The pipeline of single-view layout estimation methods (Liu et al., 2019; Yang et al., 2022; Liu et al., 2018; Stekovic et al., 2020) can be formulated as:

$$\mathcal{I} \xrightarrow{f_1} \{2D, 3D\} \xrightarrow{f_3} \{P, L, J, W\}, \tag{1}$$

where $f_1$ is a function that predicts 2D and 3D information from the input single view. Generally speaking, the final layout result $\{P, L, J, W\}$ can be directly inferred from the outputs of $f_1$. However, errors arising from $f_1$ usually adversely affect the results. Hence, a refinement step that utilizes prior information about room layout is employed to further improve the performance. Therefore, $f_3$ typically encompasses post-processing and refinement steps where the post-processing step generates an initial layout estimation, and the refinement step improves the final results.

For instance, Yang et al. (2022) chooses the HRnet network (Wang et al., 2020) as $f_1$ backbone to extract 2D plane $p$, line $l$, and predict 3D plane normal $n$ and offset $d$ from the input single view. After obtaining the initial 3D layout from the outputs of $f_1$, the method reprojects the 3D

line to a 2D line $\hat{l}$ on the image and compares it with the detected line $l$ from $f_1$. $f_3$ minimizes the error $\|\hat{l} - l\|_2^2$ to optimize the 3D plane normal. In other words, it uses the better-detected 2D line to improve the estimated 3D plane normal. In contrast, Stekovic et al. (2020) uses a different approach: its $f_1$ predicts a 2.5D depth map instead of a 2D line $l$ and uses the more accurate depth results to refine the estimated 3D plane normal. Among the works that follow the general framework of 1 (Liu et al., 2019; 2018), Yang et al. (2022) stands out as the best single-view perspective image layout estimation method without relying on the Manhattan assumption. Therefore, we present its formulation in equation (2) and extend it to multi-view scenarios.

$$\mathcal{I} \xrightarrow{f_1} \{p, l, \boldsymbol{n}, d\} \xrightarrow{f_3} \{\boldsymbol{P}, \boldsymbol{L}, \boldsymbol{J}, \boldsymbol{W}\}, \tag{2}$$

In room layout estimation from unposed multi-view images, two primary challenges aris: 1) camera pose estimation, and 2) 3D information estimation from multi-view inputs. Camera pose estimation is particularly problematic given the scarcity of annotated multi-view layout dataset. Thanks to the recent advancements in 3D vision with pretrain model, this challenge could be effectively bypassed: DUSt3R (Wang et al., 2024) has demonstrated the ability to reconstruct scenes from unposed images without requiring camera intrinsic or extrinsic, and even without overlap between views. Moreover, the 3D pointmap generated from DUSt3R can provide significantly improved 3D information, such as plane normal and offset, compared to single-view methods (Yang et al., 2022) (see Table 1 of experiment section). Therefore, DUSt3R represents a critical advancement in extending single-view layout estimation to multi-view scenarios. Before formulating the multi-view solution, we first present the key 3D representation of DUSt3R: the pointmap $\boldsymbol{X}$ and the camera pose $\boldsymbol{T}$. The camera pose $\boldsymbol{T}$ is obtained through global alignment, as described in the DUSt3R (Wang et al., 2024)).

- **Pointmap $\boldsymbol{X}$:** Given a set of RGB images $\{\mathcal{I}_1, \dots, \mathcal{I}_n\} \in \mathbb{R}^{H \times W \times 3}$, captured from distinct viewpoints of the same indoor scene, we associate each image $\mathcal{I}_i$ with a canonical pointmap $\boldsymbol{X}_i \in \mathbb{R}^{H \times W \times 3}$. The pointmap represents a one-to-one mapping from each pixel $(u, v)$ in the image to a corresponding 3D point in the world coordinate frame: $(u, v) \in \mathbb{R}^2 \mapsto \boldsymbol{X}(u, v) \in \mathbb{R}^3$.
- **Camera Pose $\boldsymbol{T}$:** Each image $\mathcal{I}_i$ is associated with a camera-to-world pose $\boldsymbol{T}_i \in SE(3)$.

Now, the sparse-view layout estimation problem can be formulated as shown in equation (3)

$$\{\mathcal{I}_1, \mathcal{I}_2, \dots\} \xrightarrow{f_1, f_2} \{p, l, \boldsymbol{X}, \boldsymbol{T}\} \xrightarrow{f_3} \{\boldsymbol{P}, \boldsymbol{L}, \boldsymbol{J}, \boldsymbol{W}\}. \tag{3}$$

In this work, we adopt the HRnet backbone from Yang et al. (2022) as $f_1$. In the original DUSt3R (Wang et al., 2024) formulation, the ground truth pointmap $\boldsymbol{X}^{obj}$ represents the 3D coordinates of the entire indoor scene. In contrast, we are interested in plane pointmap $\boldsymbol{X}^p$ that represents the 3D coordinates of structural plane surfaces, including walls, floors, and ceilings. This formulation intentionally disregards occlusions caused by non-structural elements, such as furniture within the room. Our objective is to predict the scene layout pointmap without occlusions from objects, even when the input images contain occluding elements. For simplicity, any subsequent reference to $\boldsymbol{X}$ in this paper refers to the newly defined plane pointmap $\boldsymbol{X}^p$. We introduce Plane-DUSt3R as $f_2$ and directly infer the final layout via $f_3$ without the need for any refinement.

## 3.2 $f_2$: PLANE-BASED DUST3R

The original DUSt3R outputs pointmaps that capture all 3D information in a scene, including furniture, wall decorations, and other objects. However, such excessive information introduces interference when extracting geometric primitives for layout prediction, such as planes and lines. To obtain a structural plane pointmap $\boldsymbol{X}$, we modify the data labels from the original depth map (Figure 4 (a)) to the **structural plane depth map** (Figure 4 (b)), and then retrain the DUSt3R model. This updated objective guides DUSt3R to predict the pointmap of the planes while ignoring other objects. The original DUSt3R does not guarantee output at a metric scale, so we also trained a modified version of Plane-DUSt3R that produces **metric-scale** results.

Given a set of image pairs $\mathbb{P} = \{(\mathcal{I}_i, \mathcal{I}_j) \mid i \neq j, 1 \leq i, j \leq n, \mathcal{I} \in \mathbb{R}^{H \times W \times 3}\}$, for each image pair, the model comprises two parallel branches. As shown in Figure 3, the detail of the architecture can be found in Appendix A. The regression loss function is defined as the

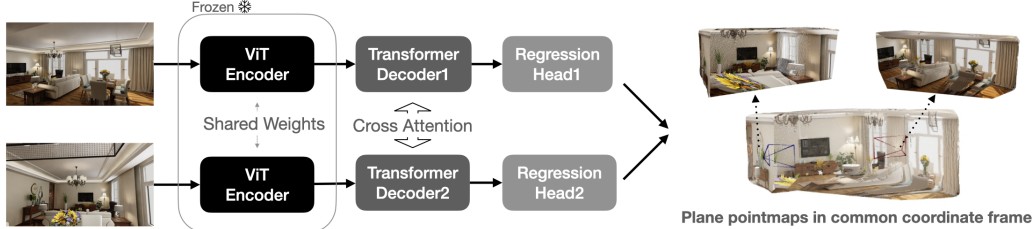

Figure 3: Plane-DUSt3R architecture remains identical to DUSt3R. The transformer decoder and regression head are further fine-tuned on the occlusion-free depth map (see Figure 4).

scale-invariant Euclidean distance between the normalized predicted and ground-truth pointmaps: $l_{regr}(v, i) = \|\frac{1}{z}\boldsymbol{X}_i^v - \frac{1}{\bar{z}}\bar{\boldsymbol{X}}_i^v\|_2^2$, where view $v \in \{1, 2\}$ and $i$ is the pixel index. The scaling factors $z$ and $\bar{z}$ represent the average distance of all corresponding valid points to the origin. In addition, by incorporating the confidence loss, the model implicitly learns to identify regions that are more challenging to predict. As in DUSt3R (Wang et al., 2024), the confidence loss is defined as: $\mathcal{L}_{\text{conf}} = \sum_{v \in \{1,2\}} \sum_{i \in \mathcal{D}^v} C_i^{v,1} \ell_{\text{regr}}(v, i) - \alpha \log C_i^{v,1}$, where $\mathcal{D}^v \subseteq \{1, \ldots, H\} \times \{1, \ldots, W\}$ are sets of valid pixels on which the ground truth is defined.

**Structural plane depth map.** The Structure3D dataset provides ground truth plane normal and offset, allowing us to re-render the plane depth map at the same camera pose (as shown in Figure 4). We then transform the structural plane depth map $D^p$ to pointmap $\boldsymbol{X}^v$ in the camera coordinate frame $v$. This transformation is given by $\boldsymbol{X}_{i,j}^v = \boldsymbol{K}^{-1}[i\boldsymbol{D}_{i,j}^p, j\boldsymbol{D}_{i,j}^p, \boldsymbol{D}_{i,j}^p]^\top$, where $\boldsymbol{K} \in \mathbb{R}^{3 \times 3}$ is the camera intrinsic matrix. Further details of this transformation can be found in Wang et al. (2024).

**Metric-scale.** In the multi-view setting, scale variance is required, which differs from DUSt3R. To accommodate this, we modify the regression loss to bypass normalization for the predicted pointmaps when the ground-truth pointmaps are metric. Specifically, we set $z := \bar{z}$ whenever the ground-truth is metric, resulting in the following loss function $l_{regr}(v, i) = \|\boldsymbol{X}_i^v - \bar{\boldsymbol{X}}_i^v\|_2^2/\bar{z}$.

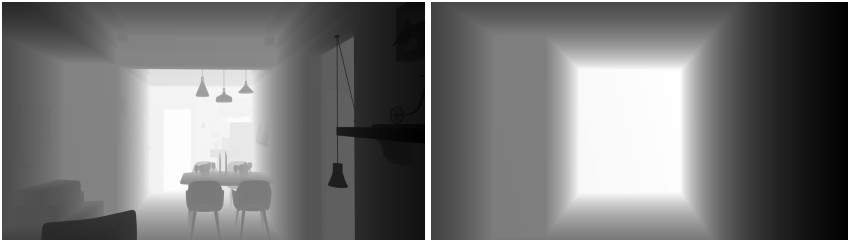

(a) The original DUSt3R depth map.   (b) The Plane-DUSt3R depth map.

Figure 4: The (a) original DUSt3R depth map and (b) occlusion removed depth map.

## 3.3 $f_3$: POST-PROCESSING

In this section, we introduce how to combine the multi-view plane pointmaps $\boldsymbol{X}$ and 2D detection results $p, l$ to derive the final layout $\{\boldsymbol{P}, \boldsymbol{L}, \boldsymbol{J}, \boldsymbol{W}\}$. For each single view $\mathcal{I}_i$, we can infer a partial layout result $\{\tilde{\boldsymbol{P}}_i, \tilde{\boldsymbol{L}}_i, \tilde{\boldsymbol{J}}_i, \tilde{\boldsymbol{W}}_i\} = g_1(\boldsymbol{X}_i, p^i, l^i)$ from the single view pointmaps $\boldsymbol{X}_i$ and 2D detection results $p^i, l^i$ through a post-process algorithm $g_1$ in camera coordinate. Then, a correspondence-establish and merging algorithm $g_2$ combines all partial results to get the final layout $\{\boldsymbol{P}, \boldsymbol{L}, \boldsymbol{J}, \boldsymbol{W}\} = g_2(\{\tilde{\boldsymbol{P}}_1, \tilde{\boldsymbol{L}}_1, \tilde{\boldsymbol{J}}_1, \tilde{\boldsymbol{W}}_1\}, \ldots)$.

**Single-view room layout estimation $g_1$.** For an image $\mathcal{I}_i$, $g_1$ mainly addresses two tasks: 1) lifting 2D planes to 3D camera coordinate space with 3D normal from pointmap $\boldsymbol{X}_i$, and 2) inferring the wall adjacency relationship. We follow the post-processing procedure in Yang et al. (2022) but with two improvements. First, the plane normal $\boldsymbol{n}$ and offset $d$ are inferred from $\boldsymbol{X}_i$ instead of directly regressed by network $f_1$. The points from $\boldsymbol{X}_i$ which belong to same plane are used to calculate $\boldsymbol{n}$ and $d$. Second, with the better 3D information pointmap $\boldsymbol{X}_i$ we can better infer pseudo wall

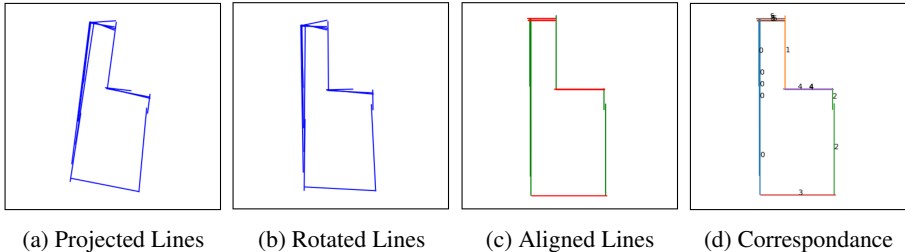

(a) Projected Lines    (b) Rotated Lines    (c) Aligned Lines    (d) Correspondance

Figure 5: (a) Planes are projected onto the x-z plane as 2D line segments. (b) The scene is rotated so that line segments are approximately horizontal or vertical. (c) Line segments are classified and aligned to be either horizontal or vertical. (d) Merged planes are shown, with segments belonging to the same plane indicated by the same color and index.

adjacency through the depth consistency of inferred plane intersection $L$ (inferred from 2D plane $p$) and predicted line region $L$ (extracted from the region of $X_i$). In our experiments, the depth consistency tolerance $\epsilon_1$ is set to 0.005.

**Multi-view room layout estimation** $g_2$. Based on the results of $g_1$, $g_2$ uses the global alignment of DUSt3R (refer to appendix A) to get the camera pose $T_i$ for each image $\mathcal{I}_i$. Then, we can transform all partial layout results ($\{\tilde{P}_1, \tilde{L}_1, \tilde{J}_1, \tilde{W}_1\}, \ldots$) to the same coordinate space. In this coordinate space, we establish correspondence for each plane, then merge and assign a unique ID to them.

Since we allow at most one floor and one ceiling detection per image, we simply average the parameters from all images to obtain the final floor and ceiling parameters. As for walls, we assume all walls are perpendicular to both the floor and ceiling. To simplify the merging process, we project all walls onto the x-z plane defined by the floor and ceiling. This projection reduces the problem to a 2D space. making it easier to identify and merge corresponding walls. Figure 5 illustrates the entire process of merging walls. Each wall in an image is denoted as one line segment, as shown in Figure 5a. We then rotate the scene so that all line segments are approximately horizontal or vertical, as depicted in 5b. In Figure 5c, each line segment is classified and further rotated to be either horizontal or vertical, based on the assumption that all adjacent walls are perpendicular to each other.

Figure 5d shows the final result after the merging process. The merging process could be regarded as a classical **Minimum Cut problem**. In Figure 5d, all line segments can be regarded as a node in a graph, two nodes have a connection if and only if they satisfy two constraints. 1) they belong to the same categories (vertical or horizontal). 2) they do not appear in the same image. 3) they are not across with the other category of node. Finally, the weight of each connection is settled as the Euclidean distance of their line segment centers. Based on this established graph, the merging results are the optimal solution of the minimum cut on this graph. The detail of the merging process can be found in Algorithm 1 of Appendix.

## 4  EXPERIMENTS

### 4.1  SETTINGS.

**Dataset.** Structured3D (Zheng et al., 2020) is a synthetic dataset that provides a large collection of photo-realistic images with detailed 3D structural annotations. Similar to Yang et al. (2022), the dataset is divided into training, validation, and test sets at the scene level, comprising 3000, 250, and 250 scenes, respectively. Each scene consists of multiple rooms, with each room containing 1 to 5 images captured from different viewpoints. To construct image pairs that share similar visual content, we retain only rooms with at least two images. Within each room, images are paired to form image sets. Ultimately, we obtained 115,836 image pairs for the training set and 11,030 image pairs for the test set. For validation, we assess all rooms from the validation set. For rooms that only have one image, we duplicate that image to form image pairs for pointmap retrieval. In the subsequent inference process, we retain only one pointmap per room.

**Training details.** During training, we initialize the model with the original DUSt3R checkpoint. We freeze the encoder parameters and fine-tune only the decoder and DPT heads. Our data augmentation

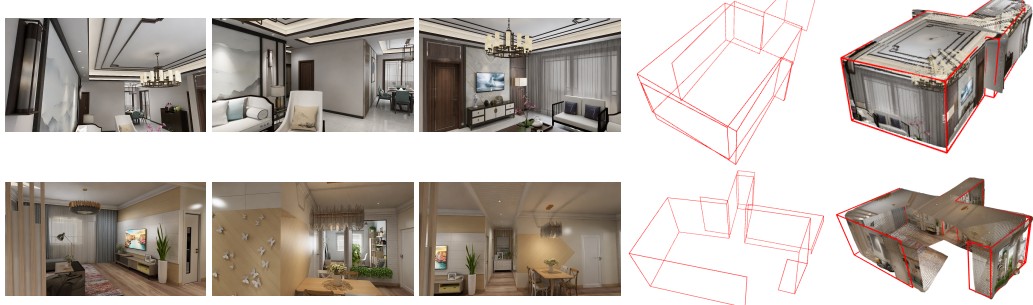

Figure 6: Qualitative comparison on the Structure3D test set. From left to right: (1-3) input views of the scene, (4) result by baseline method (Noncuboid+MASt3R) shown in wireframe only, and (5) our method's result visualized with predicted point clouds. Please refer to the appendix for more complete results.

strategy follows the same approach as DUSt3R, using input resolution of $512 \times 512$. We employ the AdamW optimizer (Loshchilov & Hutter, 2017) with a cosine learning rate decay schedule, starting with a base learning rate of 1e-4 and a minimum of 1e-6. The model is trained for 20 epochs, including 2 warm-up epochs, with a batch size of 16. We train two versions Plane-DUSt3R, one with metric-scale loss and the other one without it. We name the metric-scale one as **Plane-DUSt3R (metric)** and the other one as **Plane-DUSt3R** .

**Evaluation.** Following the task formulation in equation (3), our evaluation protocol consists of three parts to assess $f_1$, $f_2$, and the overall performance, respectively.

- For the 2D information extraction module $f_1$, we use the same metric as Yang et al. (2022) for comparison: **Intersection over Union (IoU)**, **Pixel Error (PE)**, **Edge Error (EE)**, and **Root Mean Square Error (RMSE)**.

- For the multi-view information extraction module $f_2$, we report the **Relative Rotation Accuracy (RRA)** and **Relative Translation Accuracy (RTA)** for each image pair to evaluate the relative pose error. We use a threshold of $\tau = 15$ to report RTA@15 and RRA@15 (The comprehensive results of different thresholds can be seen in Table 4 of Appendix C.1). Additionally, we calculate the **mean Average Accuracy (mAA30)**, defined as the area under accuracy curve of the angular differences at $min$(RRA@30, RTA@30).

- Finally, for evaluating the overall layout estimation, we employ **3D precision** and **3D recall** of planes as metrics. A predicted plane is considered matched with a ground truth plane if and only if the angular difference between them is less than 10°and the offset difference is less than 0.15m. Each ground truth plane can be matched only once.

**Baselines.** As this work is the first attempt at 3D room layout estimation from multi-view perspective images, there are no existing baseline methods for direct comparison. Therefore, we design two reasonable baseline methods. We also compare our $f_1$ and $f_2$ with other methods of the same type.

- Since we use Noncuboid (Yang et al., 2022) as our $f_1$, we not only compare it against the baselines from their paper (Liu et al., 2019; Stekovic et al., 2020), but also retrain it with better hyper-parameters obtained through grid search.

- For $f_2$ (Plane-DUSt3R), we compare it to recent data-driven image matching DUSt3R (Wang et al., 2024) and MASt3R (Leroy et al., 2024).

- Finally, for the overall multi-view layout baseline, we design two methods: 1) Noncuboid with ground truth camera poses and 2) Noncuboid with MASt3R. In this context, we introduce the fusion of MASt3R (Leroy et al., 2024) and NonCuboid (Yang et al., 2022) as our baseline method. MASt3R further extends DUSt3R, enabling it to estimate camera poses at a metric scale from sparse image inputs. We employ the original NonCuboid method to obtain single-view layout reconstruction. Next, we utilize the predicted camera poses to unify all planes from their respective camera poses into a common coordinate system. For instance, we designate the coordinate system of the first image as the world coordinate system. We then perform the same operation as described in Sec 3.3 to achieve the final multi-view reconstruction. The Noncuboid with ground truth camera

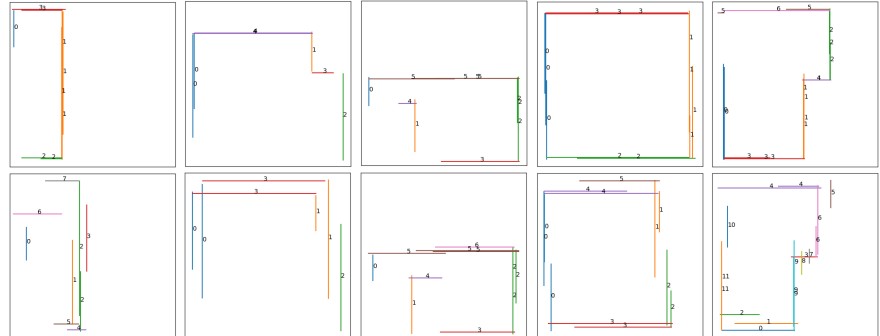

Figure 7: Birdview of multi-view 3D planes aligned to the same coordinate. The first row shows 5 cases of our pipeline results after post-processing step. The second row is the results of Noncuboid+MASt3R. Line segments of the same color indicate that they belong to the same plane.

Table 1: Quantitative results on Structure3D dataset.

| Method | re-IoU(%)↑ | re-PE(%)↓ | re-EE↓ | re-RMSE↓ | 3D-precision(%)↑ | 3D-recall(%)↑ |
|---|---|---|---|---|---|---|
| Noncuboid + MASt3R | 74.51 | 8.57 | 12.72 | 0.4831 | 37.00 | 43.39 |
| Noncuboid + GT pose | 75.93 | 7.97 | 11.37 | 0.4457 | 46.96 | **50.59** |
| Ours (metric) | 75.34 | 8.60 | 10.83 | 0.4388 | 48.98 | 45.35 |
| Ours (aligned) | **76.84** | **7.82** | **9.53** | **0.4099** | **52.63** | 48.37 |

poses is introduced as an ablation study to eliminate the effects of inaccurate pose estimation. The experimental setup is the same as the Noncuboid with MASt3R pipeline, except for the use of ground truth camera poses instead of poses estimated by MASt3R.

## 4.2 MULTI-VIEW ROOM LAYOUT ESTIMATION RESULTS

In this section, we compare our multi-view layout estimation pipeline with two baseline methods, both qualitatively and quantitatively. Additionally, we conduct experiments to verify the effectiveness of our pipeline components $f_1$ 2D detector and $f_2$ Plane-DUSt3R.

**Layout results comparison.** Table 1 and Figure 6 present quantitative and qualitative comparisons of our pipeline with two baseline methods. Ours (metric) and Ours (aligned) in Table 1 refer to the methods from our pipeline using Plane-DUSt3R (metric) and Plane-DUSt3R, respectively. The first 4 metrics (re-IoU, re-PE, re-EE, and re-RMSE) are calculated similarly to their 2D counterparts (IoU, PE, EE, and RMSE), except that the predicted 2D results are reprojected from the estimated multi-view 3D layout. Compared with the baseline methods, Plane-DUSt3R achieves superior 3D plane normal estimations compared to Noncuboid's single-view plane normal estimation, even when using ground truth camera pose (Noncuboid + GT pose). Figure 7 further demonstrates that Plane-DUSt3R could predict accurate and robust 3D information with sparse-view input.

Table 2: Comparison with data-driven image matching approaches.

| | Methods | RealEstate10K | Structured3D | | | CAD-estate | | |
|---|---|---|---|---|---|---|---|---|
| | | mAA@30 | RRA@15 | RTA@15 | mAA@30 | RRA@15 | RTA@15 | mAA@30 |
| (a) | DUSt3R (Wang et al., 2024) | 61.2 | 89.44 | 85.00 | 76.13 | 99.88 | 84.82 | 76.38 |
| | MASt3R (Leroy et al., 2024) | 76.4 | 92.94 | 89.77 | 85.34 | 99.94 | 99.00 | 95.29 |
| (b) | Plane-DUSt3R (metric) | - | 98.21 | 96.66 | 90.67 | 94.61 | 70.52 | 61.48 |
| | Plane-DUSt3R (aligned) | - | 97.95 | 96.59 | 91.80 | 94.96 | 73.74 | 64.21 |

**3D information prediction and correspondence-established method Plane-DUSt3R $f_2$.** Table 2 shows the comparison results of our Plane-DUSt3R (part (b) in Table 2), recent popular data-driven image matching approaches (part (a) in Table 2) in RealEstate10K (Zhou et al., 2018), Structure3D (Zheng et al., 2020), and CAD-Estate (Rozumnyi et al., 2023) datasets. Note that in part(b), results on RealEstate10K dataset are not provided since our model is specifically designed to predict room structural elements, while RealEstate10K contains outdoor scenes that may lead to prediction

Table 3: 2D detectors comparison on Structure3D dataset.

| Method | IoU(%)↑ | PE(%)↓ | EE↓ | RMSE↓ |
|---|---|---|---|---|
| Planar R-CNN (Liu et al., 2019) | 79.64 | 7.04 | 6.58 | 0.4013 |
| Rac (Stekovic et al., 2020) | 76.29 | 8.07 | 7.19 | 0.3865 |
| Noncuboid (Yang et al., 2022) | 79.94 | 6.40 | 6.80 | 0.2827 |
| Noncuboid (re-trained) | **80.18** | **6.13** | **6.41** | **0.2631** |

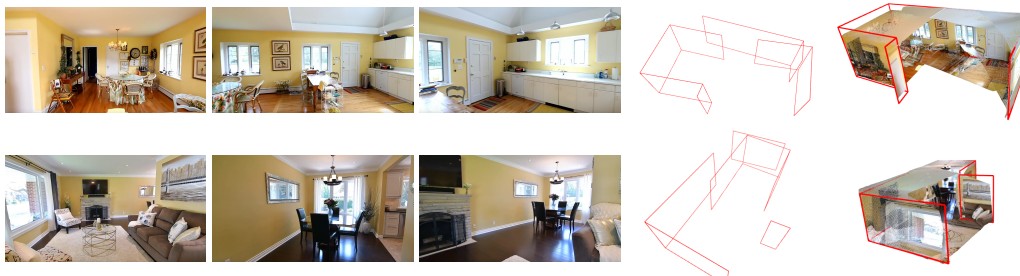

Figure 8: Qualitative comparison on in-the-wild data (Zhou et al., 2018). From left to right: (1-3) three input views of the scene, (4) layout reconstruction by baseline method (Noncuboid+MASt3R) shown in wireframe only, and (5) our method's result visualized with predicted point clouds.

failures. Instead, we utilize CAD-Estate dataset, which is derived from RealEstate10K with additional room layout annotations, as a more suitable benchmark for comparison. The results of parts (a) and (b) on three datasets show the advancements of MASt3R, not only in traditional multi-view datasets (RealEstate10K,CAD-Estate), but also in sparse-view dataset (Structure3D). Plane-DUSt3R could get a better performance compared to the previous SOTA MASt3R. One arguable point is that Plane-DUSt3R is obviously better since it is fine-tuned on Strucutre3D. That is the message we want to convey. DUSt3R/MASt3R are the SOTAs in both multi-view and saprse-view camera pose estimation tasks. After our improvements (section 3.2) and fine-tuning, Plane-DUSt3R could get 5.33 points better on the sparse-view layout dataset.

**Comparison of 2D detectors** ($f_1$). We retrain the Noncuboid method with a more thorough hyperparameter grid search, resulting in an improved version. Table 3 compares its results with other baseline methods from Yang et al. (2022).

**Comparison of various input views.** We experiment the impact of different number of input views in Appendix C.2. The results in Table 5 show a general improvement trend as the number of views increases.

### 4.3 GENERALIZABILITY TO UNKNOWN AND OUT-OF-DOMAIN DATA

Figure 1 and 12 also demonstrate the generalizability of our pipeline. It not only performs well on the testing set of Structure3D (Figure 6), but also generalizes well to new datasets, such as RealEstate10K (Figure 8 shows examples from this dataset). Furthermore, our pipeline proves effective even with data in the wild as shown in appendix in Figure 11,12. We also experiment our pipeline on the dataset CAD-estate (see Appendix E for details).

## 5 CONCLUSION

This paper introduces the first pipeline for multi-view layout estimation, even in sparse-view settings. The proposed pipeline encompasses three components: a 2D plane detector, a 3D information prediction and correspondence establishment method, and a post-processing algorithm. As the first comprehensive approach to the multi-view layout estimation task, this paper provides a detailed analysis and formulates the problem under both single-view and multi-view settings. Additionally, we design several baseline methods for comparison to validate the effectiveness of our pipeline. Our approach consistently outperforms the baselines on both 2D projection and 3D metrics. Furthermore, our pipeline not only performs well on the synthetic Structure3D dataset, but generalizes effectively to in-the-wild datasets and scenarios with different image styles such as the cartoon style.

ACKNOWLEDGMENTS

This work is partially supported by the Hong Kong Center for Construction Robotics (Hong Kong ITC-sponsored InnoHK center), the National Natural Science Foundation of China (Grant No. 62306261), CUHK Direct Grants (Grant No. 4055190), and The Shun Hing Institute of Advanced Engineering (SHIAE) No. 8115074.

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

## A    DUST3R DETAILS

Given a set of RGB images $\{I_1, I_2, \ldots, I_n\} \in \mathbb{R}^{H \times W \times 3}$, we first pair them to create a set of image pairs $\mathbb{P} = \{(I_i, I_j) \mid i \neq j, 1 \leq i, j \leq n\}$. For each image pair $(I_i, I_j) \in \mathbb{P}$, the model estimates two point maps $X_{i,i}, X_{j,i}$, along with their corresponding confidence maps $C_{i,i}, C_{j,i}$. Specifically, both pointmaps are expressed in the camera coordinate system of $I_i$, which implicitly accomplishes dense 3D reconstruction.

The model consists of two parallel branches, as shown in Fig 3, each branch responsible for processing one image. The two images are first encoded in a Siamese manner with weight-sharing ViT encoder(Dosovitskiy et al., 2020) to produce two latent features $F_1, F_2$: $F_i = \text{Encoder}(I_i)$. Next, $F_1, F_2$ are fed into two identical decoders that continuously share information through cross-attention mechanisms. By leveraging cross-attention mechanisms, the model is able to learn the relative geometric relationships between the two images. Specifically, for each encoder block:

$$
\begin{aligned}
G_{1,i} &= \text{DecoderBlock}_{1,i}(G_{1,i-1}, G_{2,i-1}), \\
G_{2,i} &= \text{DecoderBlock}_{2,i}(G_{1,i-1}, G_{2,i-1})
\end{aligned}
\tag{4}
$$

where $G_{1,0} := F_1, G_{2,0} := F_2$. Finally, the DPT(Ranftl et al., 2021) head regresses the pointmap and confidence map from the concatenated features of different layers of the decoder tokens:

$$
\begin{aligned}
X_{1,1}, C_{1,1} &= \text{Head}_1(G_{1,0}, G_{1,1}, \ldots, G_{1,B}) \\
X_{2,1}, C_{2,1} &= \text{Head}_2(G_{2,0}, G_{2,1}, \ldots, G_{2,B})
\end{aligned}
\tag{5}
$$

where $B$ is the number of decoder blocks. The regression loss function is defined as the scale-invariant Euclidean distance between the normalized predicted and ground-truth pointmaps:

$$
l_{regr}(v, i) = \| \frac{1}{z} X_i^{v,1} - \frac{1}{\bar{z}} \bar{X}_i^{v,1} \|_2^2
\tag{6}
$$

where $v \in \{1, 2\}$ and $i$ is the pixel index. The scaling factors $z$ and $\bar{z}$ represent the average distance of all corresponding valid points to the origin. The original DUSt3R couldn't guarantee output at a metric scale, so we also trained a modified version of Plane-DUSt3R that produces metric-scale results. The key change we made was setting $z := \bar{z}$. By introducing the regression loss in confidence loss, the model could implicitly learn how to identify regions that are more challenging to predict compared to others. Same as in DUSt3R (Wang et al., 2024):

$$
\mathcal{L}_{\text{conf}} = \sum_{v \in \{1,2\}} \sum_{i \in \mathcal{D}^v} C_i^{v,1} \ell_{\text{regr}}(v, i) - \alpha \log C_i^{v,1}
\tag{7}
$$

To obtain the ground-truth pointmaps $X^{v,1}$, we first transform the ground truth depthmap $D \in \mathbb{R}^{H \times W}$ into a pointmap $X^v$ express in the camera coordinate of $v$ by $X_{i,j}^v = K^{-1}[iD_{i,j}, jD_{i,j}, D_{i,j}]^\top$ with camera intrinsic matrix $K \in \mathbb{R}^{3 \times 3}$. Then we obtain $X^{v,1}$ by $X^{v,1} = T_1^{-1} T_v h(X^v)$ with $T_1, T_v \in \mathbb{R}^{3 \times 4}$ the camera-to-world poses and $h$ being the homogeneous transformation.

**Global Alignment**    For global alignment, we aim to assign a global pointmap and camera pose for each image. First, the average confidence scores of each pair of images are utilized as the similarity scores. A higher value of confidence implies a stronger visual similarity between the two images. These scores are employed to construct a Minimum Spanning Tree, denoted as $\mathcal{G}(\mathcal{V}, \mathcal{E})$, where each vertex $\mathcal{V}$ corresponding to an image in the input set and each edge $e = (n, m) \in \mathcal{E}$ indicates that images $I_n$ and $I_m$ share significant visual content. We aim to find globally aligned point maps$\{\chi^n \in \mathbb{R}^{H \times W \times 3}\}$ and a transformation $T_i \in \mathbb{R}^{3 \times 4}$ than transform the prediction into the world coordinate frame. To do this, for each image pair $e = (n, m) \in \mathcal{E}$ we have two point maps $X^{n,n}, X^{m,n}$ and their confidence maps $C^{n,n}, C^{m,n}$. For simplicity, we use the annotation $X^{n,e} := X^{n,n}, X^{m,e} := X^{m,n}$. Since $X^{n,e}$ and $X^{m,e}$ are in the same coordinate frame, $T_e := T_n$ should align both point maps with the world-coordinate. We then solve the following optimization problem:

$$
\chi^* = \arg \min_{\chi, T, \sigma} \sum_{e \in \mathcal{E}} \sum_{v \in e} \sum_{i=1}^{HW} C_i^{v,e} \| \chi_i^v - \sigma_e T_e X_i^{v,e} \|_2^2 .
\tag{8}
$$

where $v \in e$ means $v$ can be either $n$ or $m$ for the pair $e$ and $\sigma_e$ is a positive scaling factor . To avoid the trivial solution where $\sigma_e = 0$, we ensure that $\prod_e \sigma_e = 1$

# B  $f_3$ ALGORITHM

The goal of multi-view layout estimation is similar to that of single-view: we need to estimate 3D parameters for each plane and determine the relationships between adjacent planes. However, in a multi-view setting, we must ensure that each plane represents a unique physical plane in 3D space. The main challenge in multi-view reconstruction is that the same physical plane may appear in multiple images, causing duplication. Our task is to identify which planes correspond to the same physical plane across different images and merge them, keeping only one representation for each unique plane.

Since we allow at most one floor and one ceiling detection per image, we simply average the parameters from all images to obtain the final floor and ceiling parameters. As for walls, we assume all walls are perpendicular to both the floor and ceiling. To simplify the merging process, we project all walls onto the x-z plane defined by the floor and ceiling. This projection reduces the problem to a 2D space, making it easier to identify and merge corresponding walls. Figure 5 illustrates the entire process of merging walls. Each wall in an image is denoted as one line segment, as shown in Figure 5a. We then rotate the scene so that all line segments are approximately horizontal or vertical, as depicted in 5b. In Figure 5c, each line segment is classified and further rotated to be either horizontal or vertical, based on the assumption that all adjacent walls are perpendicular to each other.

---

**Algorithm 1** Merge Plane

---

**Require:** vertical lines, horizontal lines
1: Sort $verticalLines$ by x-axis value
2: Initialize $clusters$ with the first segment.
3: **for** each segment in $verticalLines[1,:]$ **do**
4:     $found \leftarrow$ False
5:     **for** each $cluster$ in $clusters$ **do**
6:         **if** $lines.image\_id$ in $cluster.image\_id$ **then**
7:             continue
8:         **end if**
9:         **if** distance($line, cluster.centroid$)$< proximity\_threshold$ **then**
10:             **if** overlap($line, cluster.centroid$)$> overlap\_threshold$ **then**
11:                 Insert $line$ into $cluster$
12:                 $found \leftarrow$ True
13:                 break
14:             **end if**
15:             **if** not intersect($line, cluster, horizontalLines, margin$) **then**
16:                 Insert $line$ into $cluster$
17:                 $found \leftarrow$ True break
18:             **end if**
19:         **end if**
20:     **end for**
21: **end for**
22: **if** $found ==$ False **then**
23:     Create a new cluster with $line$
24:     Append the new cluster to $clusters$
25: **end if**
**Ensure:** Clusters

---

# C  ADDITIONAL QUANTITATIVE RESULTS

## C.1  PERFORMANCE UNDER VARYING THRESHOLDS

To give a more complete view of the performance of the Plane-DUSt3R method, we present the results under various threshold settings (the threshold of both camera's translation and rotation) in Table 4.

Table 4: Quantitative results with different thresholds on Structure3D dataset.

| Threshold(Translation & Rotation) | 3D-precision(%)↑ | 3D-recall(%)↑ |
|---|---|---|
| 0.1m, 5° | 34.11 | 31.66 |
| 0.15m, 10° | 52.63 | 48.37 |
| 0.2m, 15° | 64.64 | 59.53 |
| 0.4m, 30° | **82.75** | **76.13** |

Table 5: Quantitative results with different input views on Structure3D dataset.

| Input views | re-IoU(%)↑ | re-PE(%)↓ | re-EE↓ | re-RMSE↓ | 3D-precision(%)↑ | 3D-recall(%)↑ |
|---|---|---|---|---|---|---|
| 2 | 75.02 | 8.72 | 8.70 | 0.4148 | 53.19 | 42.60 |
| 3 | 75.29 | 8.53 | 8.56 | 0.3596 | 54.43 | 47.97 |
| 4 | 75.55 | 8.39 | **8.55** | 0.3463 | 54.91 | 49.44 |
| 5 | **75.57** | **8.35** | 8.59 | **0.3422** | **55.02** | **49.59** |

## C.2 PERFORMANCE UNDER INPUT VIEWS

The impact of varying input views on performance is presented in Table 5. For each input views, we select this number of views from the 5 views as input and only test on rooms that have all 5 views to eliminate potential bias from room complexity variations. The results show a general improvement trend as the number of views increases.

## D ADDITIONAL QUALITATIVE RESULTS

Figure 9 showcases more visualization of our method on the Structured3D dataset, while Figure 10 presents failed cases. To demonstrate real-world applicability, we present results on in-the-wild images in Figure 11 and Figure 12.

## E EVALUATION RESULT ON CAD-ESTATE DATASET

We conducted an additional evaluation on the CAD-Estate dataset Rozumnyi et al. (2023). CAD-Estate is derived from RealEstate10K datasetZhou et al. (2018) and provides generic 3D room layouts from 2D segmentation masks. Due to differences in annotation standards between CAD-Estate and Structured3D, we selected a subset of the original data that aligns with our experimental setup. Our method and Structured3D assume a single floor, single ceiling, and multiple walls configuration. In contrast, CAD-Estate includes scenarios with multiple ceilings (particularly in attic rooms) and interconnected rooms through open doorways, whereas Structured3D treats doorways as complete walls. To ensure a fair comparison, we sampled 100 scenes containing 469 images that closely match Structure3D's annotation style. Each scene contains 2 to 10 images.

Since CAD-Estate only provides 2D segmentation annotations without 3D information, we report performance using 2D metrics: IoU and pixel error. While CAD-Estate's label classes include ["ignore", "wall", "floor", "ceiling", "slanted"], we only focus on wall, floor, and ceiling classes. We utilize the dataset's provided intrinsic parameters for reprojection during evaluation. Results are reported for both "Noncuboid + GT pose" and "Plane-DUSt3R (metric)" in Table 6. We visualize our results in Figure 13 and Figure 14

Table 6: Quantitative results with on CAD-estate dataset.

| Method | re-IoU(%)↑ | re-PE(%)↓ |
|---|---|---|
| Noncuboid + GT pose | 55.99 | 20.33 |
| Ours (metric) | **63.14** | **15.15** |

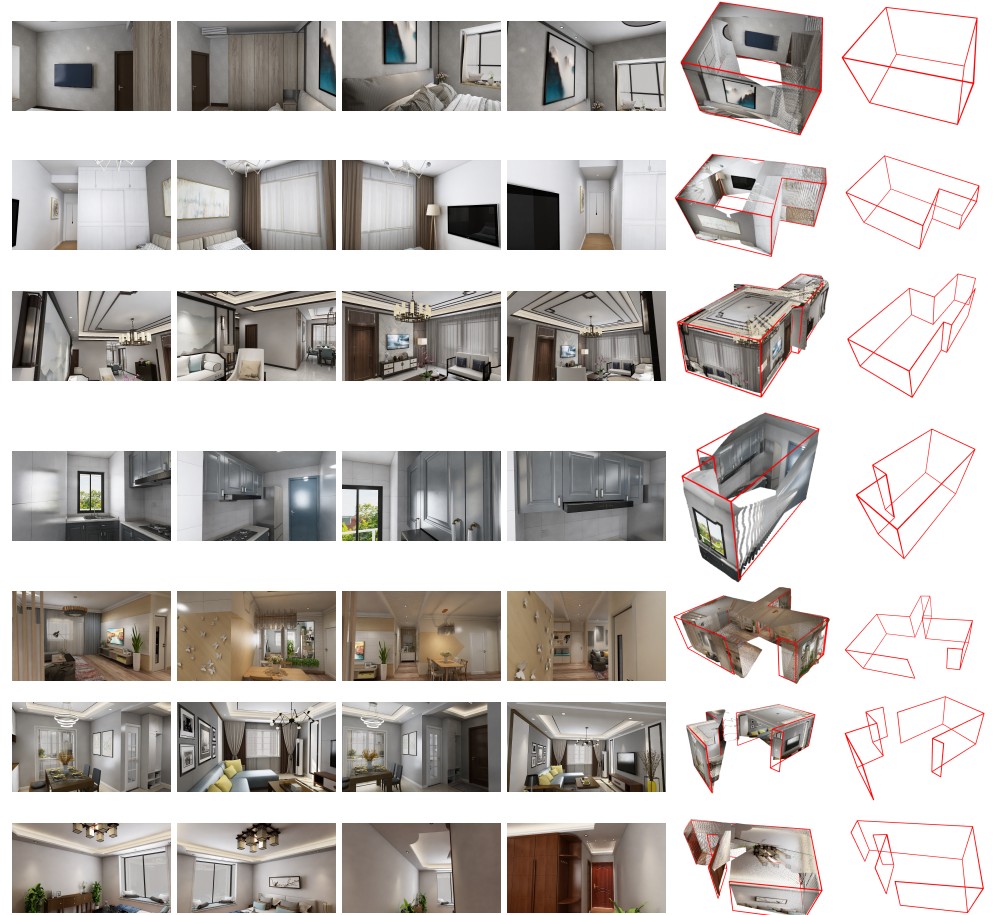

Figure 9: Qualitative results on Structure3D testing set. The 5-th column is our result visualized with pointcloud, the last column is the result shown in pure wireframe

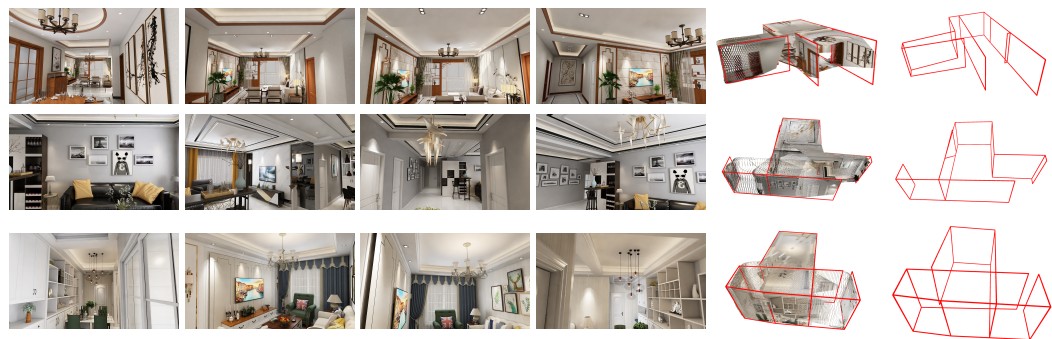

Figure 10: Failed case on Structure3D testing set. The first 4 columns are input views, the 5-th column is our result visualized with pointcloud, the last column is the result shown in pure wireframe.

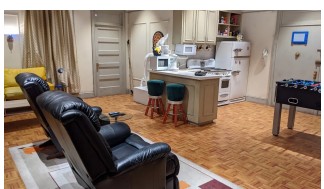 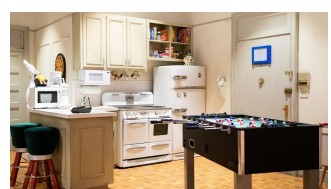 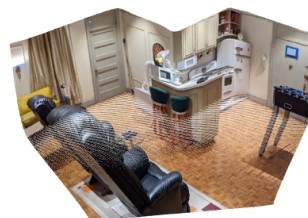

Figure 11: We provide qualitative results on in-the-wild data.

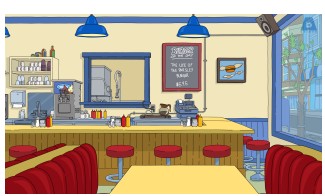 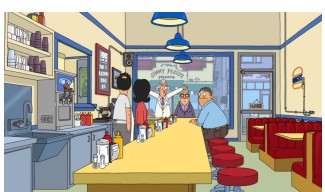 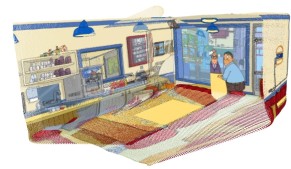

Figure 12: We provide qualitative results on out of domain cartoon data (Weber et al., 2024).

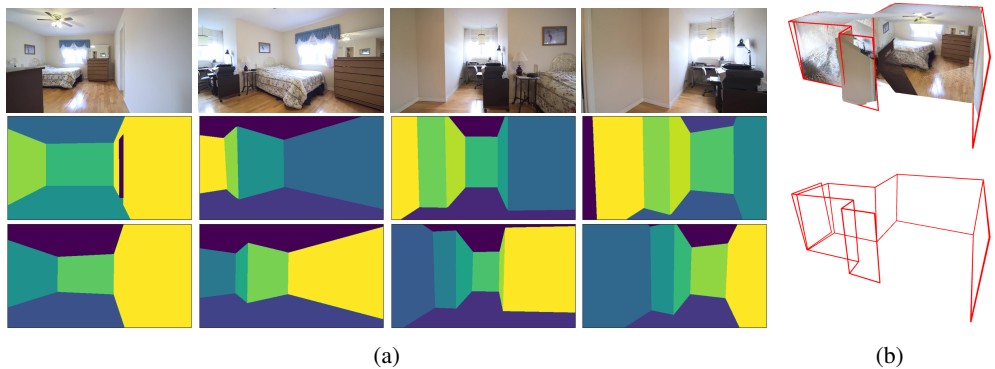

(a)                                                                              (b)

Figure 13: Visualization of results from the CAD-Estate dataset. (a) Input views are shown in the top row, followed by CAD-Estate's ground-truth segmentation in the middle row, and our predicted segmentation in the bottom row. (b) Our 3D reconstruction results displayed with point clouds (top row) and wireframe renderings (bottom row).

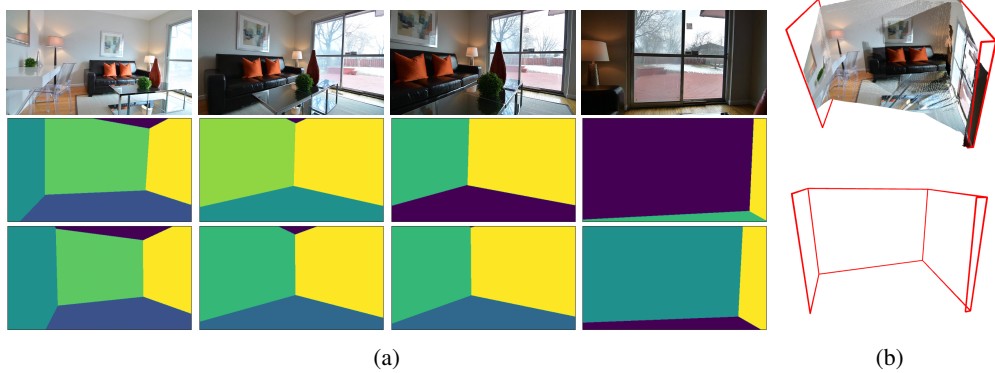

(a)                                                                              (b)

Figure 14: Visualization of results from the CAD-Estate dataset. (a) Input views are shown in the top row, followed by CAD-Estate's ground-truth segmentation in the middle row, and our predicted segmentation in the bottom row. (b) Our 3D reconstruction results displayed with point clouds (top row) and wireframe renderings (bottom row).

