# OpenReview forum: "Unposed Sparse Views Room Layout Reconstruction in the Age of Pretrain Model"
_ICLR.cc/2025/Conference — ICLR 2025 Poster_

### Official Review · Reviewer_UrzG · 2024-10-27

**Soundness:** 2
**Presentation:** 3
**Contribution:** 3
**Rating:** 6
**Confidence:** 3

**Summary:**

This work propose Plane-DUSt3R to reconstruct room layout from sparse views with unknown poses. The DUSt3R is retrained to have the amodal perception to see through the occluder and predict floors, walls, ceilings pointmap. The unknown camera parameters can then be estimated from the predicted point map. The outcome from Plane-DUSt3R is then integrated with the other prediction from single-view layout model (e.g., plane instance masks), which is then formulated as a minimum cut problem to produce the final room layout.

**Strengths:**

+ This work introduce state-of-the-art geomety prediction model, DUSt3R, into the classical layout estimation task, solving an underexplored aspect with sparse views and unknown poses.
+ The problem formulation in Sec.3.1 is clear and fluent, which unifies previous room layout estimator in a single pipeline and help readers understand how the propose method augments the existing pipeline.
+ A new benchamrk with some reasonable baselines is constructed for this task.

**Weaknesses:**

- In Table2, the performance by the proposed method on Co3Dv2 and RealEstate10K are missing. It's reasonable that DUSt3R and MASt3R perform worse as they didn't trained on the synthetic Structured3D dataset. What about the generalized performance of the proposed Plane-DUSt3R on the other real-world dataset?

- It's a pity that the proposed Plane-DUSt3R is only trained and mainly evaluated on the synthetic Structured3D dataset. There are many existing resource with multiview and room layout annotation on the 360 panorama domain, e.g., ZInD[1] has multiview layout on unfurnished rooms, MP3DLayout[2] has single-view layout annotation but the source Matterport3D also offer nearby views. Gather more data by projecting these 360 to perspective can make this work much stronger.
  - [1] Zillow Indoor Dataset: Annotated Floor Plans With 360º Panoramas and 3D Room Layouts
  - [2] Manhattan Room Layout Reconstruction from a Single 360° image: A Comparative Study of State-of-the-art Methods

- The qualitative comparison in Fig.6 and Fig.8 is unfair to the baseline. In the fourth column, the baseline "Noncuboid+MASt3R" is presented without texture. We can then easily spot one big issue that some of the walls are duplicated and misaligned. However, in the final column, the layout wireframe from the proposed method is hiding by the rgb point cloud, making it difficult to judge if the similar issue happend to the proposed method as well.

- Too few qualitative results. On the in-domain Structured3D dataset, only two are provided in Fig.6. The additional three in supp's Fig.9 also has two duplicated scenes from Fig.6. In application of room tour and room demo, the final visual outcome is what we really care about while the improvement on number is sometimes hard to interpret how it affect the final visual. In addition, some more visualization, espeically on the failure cases, may can provide some hint for future work to further improve.

**Questions:**

Seems that Sec3 only covers camera poses. Is the camera intrinsic also estimated as in DUSt3R or assumed to be known in this work?

How the accuracy vary with the number of input views? Is the performance scalable with more input views?

---

> ### Author Response · Authors · 2024-11-21
>
> Thank you for your comment. We appreciate that you found our work well-written and useful for analyzing the formulation of the multi-view layout reconstruction problem. We respond to your concern about the limited dataset, qualitative results, and technical details below.
>
> > Q1 Limitation on not enough dataset.
>
> We have conducted additional evaluations on the CAD-Estate dataset, with details, please refer to the supplementary materials in our revised version, and the overall response to all reviewers.
>
> > Q2 Qualitative results
>
> We have added more qualitative results in our revised version and inserted a column to visualize the result without texture in Fig 9 of supplementary material. We also present several failure cases in Fig 10.
>
> > Q3 Intrinsic.
>
> The intrinsic is assumed unknown and could be calculated in DUSt3r. However, during the evaluation, we used the known intrinsic data for reprojecting 3D information into 2D data.
> .
> > Q4 Performance on different input views.
>
> The impact of varying input views on performance is presented in Table 1. We align the depth based on the ground-truth relative pose and the predicted pose for multi-view cases but use a predefined scale for single-view cases. So we can observe a reduced performance for single-view setting.
> Regarding scalability, our answer is yes.  We could handle more input views. This is also demonstrated in the additional CAD-estate dataset evaluation where scenes contain 2-10 views. Since NonCuboid is run parallelly on each input image. The only bottleneck for using more images like more than 20, is that DUSt3R may go out of memory in the global alignment step. But this problem has been solved on its following work Mast3r.
>
> Table 1
> | Input View Number | IoU (%)↑ | PE (%)↓ | EE↓ | RMSE↓ | Precision (%)↑ | Recall (%)↑ |
> |------------------|-----------|----------|------|--------|----------------|-------------|
> | 1 | 68.53 | 12.10 | 27.66 | 1.6430 | 15.78 | 14.62 |
> | 2 | 78.81 | 7.18 | 12.88 | 0.3584 | **55.16** | **52.76** |
> | 3 | **78.92** | 7.09 | 10.33 | 0.5450 | 51.80 | 47.75 |
> | 4 | 78.78 | **6.98** | 9.56 | 0.4207 | 55.20 | 52.09 |
> | 5 | 75.57 | 8.35 | **8.59** | **0.3422** | 55.02 | 49.59 |

---

> ### Author Response · Authors · 2024-11-25
>
> Dear Reviewer,
>
> Thank you for your detailed review, which would help improve our paper a lot! We have carefully addressed each of your comments and concerns with additional experiments and detailed explanations. As the discussion period is nearing its end, we would greatly appreciate if you could review our responses and consider raising your score if you find that we have adequately addressed your concerns.
>
> Thank you for your time.
>
> Sincerely,
>
> Authors

---

> ### Author Response · Authors · 2024-11-27
>
> Dear reviewer,
>
>
> We thank you for the valuable time and effort in reviewing our paper.
>
> In the rebuttal, we have added the evaluation on the real-world dataset CAD-Estate in Appendix E and more qualitative results in Appendix D. Moreover, we have performed additional comparative experiments with different input views, and also explained the implementation detail about Intrinsic settings.
>
> We hope you might find our responses and revisions satisfactory, and sincerely hope you will reconsider your rating based on our clarification in responses and the revised paper. Thank you for your time!
>
> Best Regards,
>
> Authors

---

> ### Author Response · Authors · 2024-11-30
>
> Dear Reviewer,
>
> Thank you very much for your time and effort. Since the rebuttal period has reaching to the end, could you take a look at our response and reconsider your score for our submission.
>
> Best regards
>
> Authors

---

> ### Comment · Reviewer_UrzG · 2024-11-30
>
> Sorry for late reply. I appreciate authors effort in preparing the additional experiments and paper revision. Some concerns remain.
>
> **Q1**. Thanks for the evaluation on additional CAD-estate dataset. My question is: why the proposed method can not be evaluated on Co3Dv2 and RealEstate10K datasets. Please remind me if I miss some discussions or explanations. I think the comparisons on Co3Dv2 and RealEstate10K is more important as they are the real-world datasets with flourished baselines, which helps us better assess the performance comparing to the state of the art.
>
> **Q4**. More discussion for the experimental results with different number of input views is needed. We observe consistent improvement of EE and RMSE with more input views but the other four metrics are not. It seems that the IoU, PE, and 3D recall drops a lot from 4 input views to 5 input views. The author should provide more discussion about these results.

---

> > ### Comment · Reviewer_6JGC · 2024-11-30
> >
> > I'd like to comment that CAD-Estate dataset is part of the RealEstate10k dataset. CAD-Estate is the name of structural elements and object annotations on the RE10k dataset.

---

> ### Author Response · Authors · 2024-12-01
>
> Dear reviewers, thank you for your valuable comments.
>
> > Q1. Regarding Dataset Selection:
>
> We appreciate reviewer 6JGC's clarification about CAD-Estate's relationship with RealEstate10K. While RealEstate10K is indeed a valuable real-world dataset, it lacks ground-truth structural information. This is why we utilize CAD-Estate, which provides the necessary structural annotations on RealEstate10K scenes.
>
> Regarding Co3Dv2,  it's an dataset for single-object reconstruction and camera pose estimation, it's not suitable for our task of room-level layout reconstruction. The dataset primarily focuses on single objects captured from different viewpoints rather than indoor scenes with structural elements. We initially included these results to demonstrate the pose estimation capabilities of DUSt3R. As reviewer H8Ec also raised the same concern about Table2 in our paper, we could remove the comparsion of Co3Dv2 dataset in tabel2 in the camera-ready version.
>
> > Q4. Performance Across Different View Numbers:
>
> Our previous presentation might have been misleading as it included rooms with varying numbers of available views. And the most complicated cases usually falls in the 5-views settiing. We have analyzed the distribution of views across rooms in Structure3D:
>
> Table 1
> | Input View Number |  Room Number |
> |----------------|----------------|
> | 1  | 102  |
> | 2  | 171  |
> | 3  | 274  |
> | 4 | 385  |
> | 5  | 720  |
>
> For a fair comparison, we now report results only on rooms that have all 5 views available in Tabel 2, allowing us to evaluate the same rooms across different view settings. This approach eliminates potential bias from room complexity variations. The results show a general improvement trend as the number of views increases.
>
> Table 2
> | Input View Number |  IoU (%)↑ | PE (%)↓ |  EE↓ |RMSE↓|Precision (%)↑|Recall (%)↑|
> |----------------|----------------|----------------|----------------|----------------|----------------|----------------|
> | 2  | 75.02         | 8.72      | 8.70         | 0.4148       | 53.19       | 42.60 |
> | 3  | 75.29         | 8.53      | 8.56       | 0.3596       | 54.43 |  47.97 |
> | 4 |  75.55         | 8.39       | **8.55** | 0.3463       | 54.91       | 49.44 |
> | 5 | **75.57**  | **8.35** | 8.59       | **0.3422** | **55.02**       | **49.59** |

---

> ### Author Response · Authors · 2024-12-02
>
> Dear Reviewer,
>
> Thank you very much for your time and effort. Since the rebuttal period has reaching to the end, could you take a look at our response and reconsider your score for our submission.
>
> Best regards
>
> Authors

---

> > ### Comment · Reviewer_UrzG · 2024-12-02
> >
> > **Q1**. I agree. Co3Dv2 results on Table 2 is very confusing with all the baselines numbers but without the one by the proposed method. I also suggest removing them. Regarding the RealEstate10K in Table2, the proposed method should be able to report the mmA@30? It's totally fine to me if the number do not outperform the others given the additional ability to recover room layout. It's good for future work to know if there is still any aspects can still be further improved. If for some reasons the proposed method can not be evaluated on RealEstate10K, I also suggest remove the RealEstate10K column to prevent confusion.
> >
> > **Q4**. Thanks for the new evaluations. The results make sense to me now.
> >
> > I will increase my rating and lower my confidence.

---

> > > ### Author Response · Authors · 2024-12-02
> > >
> > > Thank you for your positive and constructive feedbacks! We will remove the Co3Dv2 baseline results from Table 2 as suggested to avoid confusion. And we will run experiments on RealEstate10K to evaluate pose estimation (mmA@30) in our final version.

---

### Official Review · Reviewer_oARR · 2024-11-03

**Soundness:** 3
**Presentation:** 3
**Contribution:** 3
**Rating:** 5
**Confidence:** 3

**Summary:**

This paper introduces Plane-DUSt3R, a method that leverages the DUSt3R framework to estimate structural planes from multi-view images by finetuning on a room layout dataset (Structure3D). This approach allows for room layout estimation with just a single post-processing step and 2D detection results, handling multiple-perspective images. Experimental results show that Plane-DUSt3R outperforms existing methods on synthetic datasets and is robust across various real-world image styles, including cartoons.

**Strengths:**

1. The major strength of this paper is the first time addressing 3D room layout estimation from multi images by using large reconstruction model s (i.e., DUST3R).
2. To support using DUST3R, this paper proposed Plane-DUSt3R to estimate structural planes from multi-view images by only require a single post-processing step.
3. Plane-DUSt3R achieves the best performance comparing with the baselines.

**Weaknesses:**

1. I think the major contribution of the performance boost should go to Dust3R architecture and its pretrained weight, as it is a strong prior model that gives stable and faithful 3D points and camera poses output for Plane-DUSt3R.

2. The key model of this paper is only the post-processing step of extracting layout planes, which is, technically speaking, quite incremental.

**Questions:**

None

---

> ### Author Response · Authors · 2024-11-21
>
> Thank you for your comment and your recognition of our performance in multi-view layout estimation.
>
> While we acknowledge that our method might be technically incremental, our main contribution lies in introducing and analyzing the multi-view room layout estimation problem, which was largely unexplored in previous work. We provided the first systematic study of this setting, including comprehensive baselines and analysis that will benefit future research. Our technical contribution goes beyond simply combining DUSt3R with existing single-view methods - we designed specific strategies to effectively leverage multi-view geometric constraints while maintaining robustness. The robust performance across both synthetic and real-world scenarios demonstrates the practical value of our approach. Through this work, we aimed to establish a foundation for future research in multi-view room layout estimation.

---

> ### Author Response · Authors · 2024-11-25
>
> Dear Reviewer,
>
> Thank you for your detailed review of our paper. We have carefully addressed your feedback regarding the contribution and technical aspects of our work. As the discussion deadline is approaching, we would greatly appreciate it if you could review our response and reconsider your score if you find that we have adequately addressed your concerns.
>
> Thank you for your time.
>
> Best regards,
>
> Authors

---

> ### Author Response · Authors · 2024-11-27
>
> Dear Reviewer,
>
> We thank you for your valuable time and effort in reviewing our paper.
>
> In the rebuttal, we have clarified that while individual components of our method build upon existing work, our contribution is significantly more comprehensive. We have presented the first systematic study of multi-view room layout estimation. In addition to our novel method, we have also developed two new baseline approaches from scratch, as no existing implementations were available for this specific task. These baselines provide valuable benchmarks for future research. Moreover, we have added additional evaluation on a real-world dataset which further demonstrates that our method is robust across different scenarios. We believe our work not only advances the baseline but also opens up new research directions in room layout estimation that will facilitate future research in this field.
>
> We hope you find our responses and revisions satisfactory, and sincerely hope you will reconsider your rating based on our clarifications in responses and the revised paper. Thank you for your time!
>
> Best Regards,
>
> Authors

---

> ### Author Response · Authors · 2024-11-30
>
> Dear Reviewer,
>
> Thank you very much for your time and effort. Since the rebuttal period has reaching to the end, could you take a look at our response and reconsider your score for our submission.
>
> Best regards
>
> Authors

---

> ### Author Response · Authors · 2024-12-02
>
> Dear Reviewer,
>
> Thank you very much for your time and effort. Since the rebuttal period has reaching to the end, could you take a look at our response and reconsider your score for our submission.
>
> Best regards
>
> Authors

---

### Official Review · Reviewer_H8Ec · 2024-11-04

**Soundness:** 3
**Presentation:** 3
**Contribution:** 2
**Rating:** 6
**Confidence:** 4

**Summary:**

In this work authors propose sparse-view layout reconstruction pipeline that combines existing single view layout methods such as [1] and sparse view 3D reconstruction method Dust3R [2], to generate more accurate 3D layout from sparse images.

Authors retrained [1] to more accurately detect 2D premitives. They also retrain Dust3R, to only predict layout-plane pointmaps, essentially ignoring foreground furniture. They call this new model Plane-Dust3R. Authors train their PlaneDust3R method on synthetic multi-view Structure3D dataset. And compare accuracy of multiple components of their pipeline, on this synthetic dataset.

From 2D primitives and 3D plane pointmaps, authors design post-processing pipeline that converts them to 3D plane equations, and their relationships, resulting in full 3D layout representations.


[1] Yang, Cheng, et al. "Learning to reconstruct 3d non-cuboid room layout from a single rgb image." Proceedings of the IEEE/CVF Winter Conference on Applications of Computer Vision. 2022.

[2] Wang, Shuzhe, et al. "Dust3r: Geometric 3d vision made easy." Proceedings of the IEEE/CVF Conference on Computer Vision and Pattern Recognition. 2024.

**Strengths:**

Proposed sparse multi-view layout method is first of its kind. Authors make use of existing methods and retrain them to make them work for their task.

Overall pipeline is shown to work better than naively combining existing methods such as NonCuboid + Mast3R / NonCuboid + GT Pose.

Overall paper is well written, is easy to follow. There are many components in the pipeline, but authors explain them in an understandable manner.

**Weaknesses:**

The big limitation of method is that it is only shown to work on synthetic dataset. While availability of real multi-view layout dataset is scarse, there exists many multi-view 3D datasets (such as ScanNet, ARKit Scene) etc. These datasets have ground-truth depth maps and camera poses that can be used for pseudo ground-truth generation and the proposed method can be compared against other layout reconstruction methods on such dataset. Authors do provide qualitative results on some real-world data, but it is not enough to assess accuracy of method on the real data. If authors can show effectiveness of the method on real data this method can be promising.

Second, I am not convinced as to why PlaneDust3R is needed. If authors take regular Dust3R and use off-the-shelf semantic segmentation to segment walls, floors, and ceilings and use only those points to generate layout (by combining this with NonCuboids + f3 post-processing proposed in the paper), that might also be enough. Retraining method only on synthetic dataset such as Structure3D might reduce their accuracy on the real-world dataset.

**Questions:**

In Table-2, other datasets results are not useful since this method does not have numbers for those. I would suggest authors to keep results for only structure 3D.

3D precision and recall should be reported at different threshold. I think 15deg threshold is pretty high. Maybe authors should show results at 5deg, 15deg, and 30deg angular thresholds, and similar translation thresholds. This will give more comprehensive view of the methods performance.

---

> ### Author Response · Authors · 2024-11-21
>
> Thank you for your comment. We appreciate that you found our work well-written and useful for analyzing the formulation of the multi-view layout reconstruction problem. We respond to your questions about limitations on dataset,  architecture choice, and evaluation metrics below.
>
> > Q1 Limitation on not enough dataset.
>
> A: We have conducted additional evaluations on the CAD-Estate dataset,  with details, please refer to the supplementary materials in our revised version, and the overall response to all reviewers.
>
> > Q2 Choosing PlaneDust3R rather than adding semantic segmentation to Dust3R.
>
> A: We choose this design for two main reasons. First for pipeline Efficiency. Adding semantic segmentation as a post-processing step would introduce an additional sub-task into the pipeline. Each additional processing step could potentially introduce and accumulate errors. Second is that our design philosophy aligns with the evolution of single-view layout estimation methods, which have progressed from relying on visible wall patches to directly targeting room-structural planes. This evolution has proven effective for several reasons, which are also stated in [1]: 1. Better handling of occlusions: The method can infer room planes even when they are heavily occluded by furniture. 2. More robust layout estimation: Direct plane estimation is less affected by scene clutter and partial observations.
>
> [1]Howard-Jenkins, et al. "Thinking Outside the Box: Generation of Unconstrained 3D Room Layouts." Computer Vision--ACCV 2018: 14th Asian Conference on Computer Vision. 2019
>
> > Q3 3D precision and recall under different thresholds.
>
> A: We acknowledge the error in our paper regarding the angle threshold. The correct threshold we used for calculating the metric is 10 degrees instead of 15, while the translation threshold remains at 0.15m. Table 1 presents our results under various threshold settings.
>
> Tabel1:
> | Threshold(Rotation & Translation) | Precision (%)↑ | Recall (%)↑ |
> |-----------|---------------|-------------|
> | 5° / 0.1m | 34.11 | 31.66 |
> | 10° / 0.15m | 52.63 | 48.37 |
> | 15° / 0.2m | 64.64 | 59.53 |
> | 30° / 0.4m | 82.75 | 76.13 |

---

> ### Author Response · Authors · 2024-11-25
>
> Dear Reviewer,
>
> Thank you for your detailed review, which will significantly improve our paper. We have carefully addressed each of your comments and concerns with additional experiments and detailed explanations. As the discussion period is ending soon, we would greatly appreciate if you could review our responses and reconsider your score if you find that we have adequately addressed your concerns.
>
> Thank you for your time.
>
> Sincerely,
>
> Authors

---

> > ### Comment · Reviewer_H8Ec · 2024-11-25
> > **Thank you for the rebuttal!**
> >
> > I thank authors for providing rebuttal that answers my concerns. The paper is a good addition to Dust3r family of method. I will raise my score to 6.

---

> > > ### Author Response · Authors · 2024-11-26
> > >
> > > Thank you for your positive feedbacks. Your insightful and constructive comments really help us improve the quality of our work.

---

### Official Review · Reviewer_6JGC · 2024-11-09

**Soundness:** 4
**Presentation:** 4
**Contribution:** 4
**Rating:** 10
**Confidence:** 4

**Summary:**

The paper proposes an extension of DUSt3R to room layout reconstruction by retraining and using additional processing steps. Results demonstrate superior performance.

**Strengths:**

- The method is sound with great results outperforming other methods.
- This is the first method for unposed sparse view room layout reconstruction, especially when the views are not overlapping.
- The whole idea is interesting, especially to use structural plane depth map and metric scale for DUSt3R.
- The evaluation is great, especially the design of baselines.
- The paper is well-written and was a pleasure to read.

**Weaknesses:**

- Evaluation on more datasets would be more interesting, e.g. CAD-Estate dataset with structural elements annotations.

**Questions:**

- Figures seem to have low resolution since they are most likely represented as jpg/png images. I recommend using vector graphics instead for better visualisations.

---

> ### Author Response · Authors · 2024-11-21
>
> We really appreciate your positive evaluation score of 10, which acknowledges the contributions of our work.
>
> We have conducted additional evaluations on the CAD-Estate dataset, with details, please refer to the supplementary materials in our revised version, and the overall response to all reviewers.
>
> While we would prefer to use vector graphics throughout, OpenReview's file size limitations constrained us. But we have replaced the key architectural figures with higher resolution.

---

> > ### Comment · Reviewer_6JGC · 2024-11-27
> >
> > Thank you for providing additional experiments on real data. This is very important. It makes the paper much stronger. I keep my strong accept rating.
> >
> > One thing to note is that it's not true that CAD-Estate has only 2D annotations for structural elements. There are also annotations for 3D planes and their spatial extents. I believe there are also ground truth meshes for all elements, e.g. walls, floor, ceiling. Therefore, it should be possible to have also some 3D evaluation metrics.

---

> > > ### Author Response · Authors · 2024-11-30
> > >
> > > Thank you again for your strong acknowledgment of our method and for maintaining the "strong accept" rating. Thank you for your comments, we have now updated our overall response and included 3D evaluation metrics in the final result. Your constructive feedback has helped us a lot.

---

### Author Response · Authors · 2024-11-21

Dear Reviewers,

We appreciate your valuable feedback and your recognition of our work’s novelty and performance. We noticed that there are common concerns about only training and evaluating on one synthetic dataset, and a desire to see quantitative results on some real-world data. We have carefully considered your suggestion and conducted additional experiments on the CAD-Estate dataset. Our choice of CAD-Estate is motivated by several reasons: 1. Data Quality and Scale: CAD-Estate provides room layouts from 2246 videos, offering a larger and more diverse evaluation base compared to alternatives like ScanNet layout [1] (293 views). 2. Multi-view Completeness: 360-panorama datasets primarily offer rotational variations, while CAD-Estate provides multiple views with both rotational and translational differences for each scene. 3. Practical Considerations. CAD-Estate is already well-established and has a much more similar annotation style to Structured3d. We believe the evaluation results conducted on this dataset are representative.

For evaluation settings, despite CAD-Estate's similar annotation style, some notable differences exist. Our method and Structured3D operate under the assumption of a single floor, single ceiling, and multiple walls configuration. CAD-Estate, however, presents more complex scenarios, including multiple ceiling levels (particularly in attic rooms) and interconnected rooms through open doorways (whereas Structured3D treats doorways as complete walls). To ensure a fair comparison, we carefully selected a subset of CAD-Estate data that aligns with  Structured3D's annotation style. Our final evaluation dataset consists of 100 scenes containing 469 images, with each scene containing 2 to 10 images.

We report performance using both 2D metrics (IoU and pixel error) and 3D metrics (precision and recall). While CAD-Estate's label classes include ["<ignore>", "wall", "floor", "ceiling", "slanted"], we only focus on wall, floor, and ceiling classes. We utilize the dataset's provided intrinsic parameters for reprojection during the evaluation. Results are reported for both "Noncuboid + GT pose" and "Plane-DUSt3R (metric)". Please refer to our supplementary materials for qualitative results.

| Method |re-IoU (%)↑ | re-PE (%)↓ |3D-Precision (%)↑ |3D-Recall (%)↑ |
|--------|----------|----------|---------|---------|
| Noncuboid + GT pose on cad-estate | 55.99 | 20.33 |15.59|**30.28**|
| Ours (metric) on cad-estate | **63.14** | **15.15** | **22.58**|26.55|


Our method achieves performance in re-loU (63.14% vs 55.99%) and re-PE (15.15% vs 20.33%), along with higher 3D precision (22.58% vs 15.59%) at 15 degrees and 0.2m thresholds. The relatively low precision scores can be attributed to our model's bias from training on Structured3D, where most adjacent walls are predominantly orthogonal and ceilings are typically horizontal. In contrast, CAD-Estate contains more diverse architectural features, including non-orthogonal walls and slanted ceilings. While the baseline method shows higher 3D recall, this is likely due to its tendency to generate duplicated planes and its utilization of ground-truth pose information. Additionally, some performance discrepancies may be influenced by some annotation inconsistencies in CAD-Estate's 3D triangle meshes.

[1] https://github.com/vevenom/ScanNet-Layout

---

### Meta-Review · Area_Chair_xk66 · 2024-12-19

**Metareview:**

This paper introduces a pipeline for multi-view room layout estimation, leveraging the 3D foundation model DUSt3R. Experiments on both synthetic and real-world datasets demonstrate its improved performance and generalization.

Main strengths of this paper are as below:

- The paper presents a new pipeline for multi-view room layout estimation, simplifying traditionally complex multi-view geometry into a single-step solution using the 3D foundation model DUSt3R.

- Experiments on synthetic and real-world datasets validate its improved performance and generalization.

- The paper is well-organized and offers comprehensive experimental results, making it accessible and providing valuable insights.

Needing more experiments were raised by the reviewers. Also, the new part different from the previous methods need more detailed depictions. Please revise the paper according to the discussions before submitting the final version.

**Additional Comments On Reviewer Discussion:**

- Reviewer 6JGC suggested evaluating the method on the CAD-Estate dataset, which includes 3D plane annotations and meshes. The authors responded by conducting additional evaluations on the CAD-Estate dataset and including 3D evaluation metrics in the revised version. They also replaced key architectural figures with higher-resolution versions.
- Reviewer H8Ec raised concerns about the method being evaluated only on synthetic data, suggesting the use of real-world datasets like ScanNet for comparison. The authors responded by adding real-world data evaluations and revising Table-2 with 3D precision/recall metrics at multiple thresholds. Additionally, Reviewer H8Ec questioned the necessity of Plane-DUSt3R, proposing that regular DUSt3R with off-the-shelf semantic segmentation and NonCuboids + f3 post-processing might be sufficient. The authors justified their method by highlighting the specific improvements and innovations of Plane-DUSt3R in the revised version.
- Reviewer oARR considered that the main contribution of the paper to be the post-processing step for extracting layout planes, considering it technically incremental. The authors responded by clarifying that their contribution is the first systematic study of multi-view room layout estimation, introducing new baselines, and demonstrating robust performance across synthetic and real-world datasets.
- Reviewer UrzG raised concerns about dataset selection, qualitative results, and performance across varying input views. The authors added evaluations on the CAD-Estate dataset, clarified dataset limitations, provided more qualitative results (including failure cases), and discussed performance variations with input views. They agreed to revise Table 2 for clarity.

---

### Decision · Program_Chairs · 2025-01-22

Accept (Poster)